# Effect of snowfall on changes in relative seismic velocity measured by ambient noise correlation

Antoine Guillemot[1], Alec van Herwijnen[2], Eric Larose[1], Stephanie Mayer[2], Laurent Baillet[1]

[1] Laboratoire ISTerre, Univ. Grenoble Alpes, CNRS, Univ. Savoie Mont Blanc, 38000 Grenoble, France
[2] WSL Institute for Snow and Avalanche Research SLF, Davos, Switzerland

*Correspondence to*: A. Guillemot (antoine.guillemot@univ-grenoble-alpes.fr)

**Abstract.** In mountainous, cold temperate and polar sites, the presence of a snow cover can affect relative seismic velocity changes (dV/V) derived from ambient noise correlation, but this relation is relatively poorly documented and ambiguous. In this study, we analyzed raw seismic recordings from a snowy flat field site located above Davos (Switzerland), during one entire winter season (from December 2018 to June 2019). We identified three snowfall events with a substantial response of dV/V measurements (drops of several percent between 15 and 25 Hz), suggesting a detectable change in elastic properties of the medium due to the additional fresh snow. To better interpret the measurements, we used a physical model to compute frequency dependent changes in the Rayleigh wave velocity computed before and after the events. Elastic parameters of the ground subsurface were obtained from a seismic refraction survey, whereas snow cover properties were obtained from the snow cover model SNOWPACK. The decrease in dV/V due to a snowfall were well reproduced, with the same order of magnitude as observed values, confirming the importance of the effect of fresh and dry snow on seismic measurements. We also observed a decrease in dV/V with snowmelt periods, but we were not able to reproduce those changes with our model. Overall, our results highlight the effect of the snowcover on seismic measurements, but more work is needed to accurately model this response, in particular for the presence of liquid water in the snowpack.

## 1 Introduction

The method of seismic ambient noise correlation is broadly used to monitor the subsurface, in order to detect physical processes in the surveyed medium such as changes in rigidity, fluid injection or cracking (Sens-Schönfelder and Wegler, 2006; Larose et al., 2015). Several observables such as relative velocity changes of surface waves, or changes in waveforms, can be continuously measured. These indicators can be precursors for catastrophic events such volcanic eruptions (Brenguier et al., 2008; Rivet et al., 2015) or landslides failure (Le Breton et al. 2020, for a review).

Relative seismic velocity changes (dV/V) can be estimated from daily or hourly seismic ambient noise cross-correlations, assuming (at least partially) both temporal and spatial stability of the sources (Hadziioannou et al. 2009). As the coda part of cross-correlations is mostly controlled by surface waves and scattering (Obermann et al., 2013), dV/V can be estimated in different frequency bands, corresponding to different depths of investigation (Mainsant et al. 2012, Voisin et al. 2016). Velocity changes are sensitive to environmental influences in the shallow subsurface, such as temperature (Tsai, 2011;

Richter et al., 2014; Hillers et al., 2015), atmospheric fluctuations (Hillers et al., 2015; Gradon et al., 2021), freezing-thawing
(Gassenmeier et al., 2015; James et al., 2017; Miao et al., 2019; Guillemot et al., 2020; Steinmann et al., 2021) and ground
water level fluctuations (Meier et al. 2010, Mainsant et al. 2012, Hillers et al. 2014, Rivet et al. 2015, Voisin et al. 2017,
Planès et al. 2017, Wang et al. 2017, Clements & Denolle 2018). These latter environmental effects on dV/V have been
studied both experimentally and numerically (Berger, 1975; Tsai, 2011), and have been recently reviewed in a context of
landslide monitoring (Le Breton et al., 2020). In polar and cold temperate regions, significant dV/V variations were observed
related to the presence of snow (Hotovec-Ellis et al., 2014; Wang et al., 2017). Some observations show a positive
correlation between snow depth and dV/V measurements at seasonal scale (Hotovec-Ellis et al., 2014; Wang et al., 2017),
whereas Wang et al. (2017) and Le Breton (2019, Fig. A-11) mentioned a negative correlation during intense snowfalls. In
ice sheets, Mordret et al. (2016) modelled the effect of snow accumulation in  by using poroelasticity and viscoelasticity at
seasonal scale. But to the best of our knowledge, the  effect of snow on dV/V in snowy temperate regions has not been
properly studied with high resolution (Larose et al., 2015, Fig. 10).
Snow is a highly porous material with low density and low elastic modulus (Gerling et al., 2017) Typical densities for a
seasonal snow cover range from 50 to 500 kg/m3 (Schweizer and Jamieson, 2003). Fresh snow generally has a density
between 50 and 150 kg/m3, yet due to snow settlement (compaction), density rapidly increases. Snow is a material that exists
very close to its melting point, causing rapid micro-structural changes (e.g. Herwijnen and Miller, 2013). During the winter
season, when air temperature mostly remains below freezing, there is no liquid water in the snowpack and snow
temperatures are below zero. This is called a dry snowpack. In spring, warm temperatures and solar radiation cause daily
surface melting. As a result, snowpack temperatures gradually increase to zero degrees, and the liquid water content
increases. This is called a wet snowpack.  Elastic wave velocities in snow, like most of its mechanical properties, including
the elastic modulus, are highly dependent on snow density, temperature and liquid water content. While the effect of snow
density and temperature are well documented (e.g. Schweizer and Camponovo, 2002; Sayers, 2021), the influence of liquid
water content is still poorly understood. Modelling snow acoustics is highly challenging, since acoustic phase velocities of
this porous medium strongly depends on porosity, stiffness and density of the bulk frame. Recent studies address this
dependency using rigid-frame and Biot's models, assuming pore space to be air-filled (Capelli et al., 2016; Sidler, 2015;
Sayers, 2021). Furthermore, the presence of liquid water, and with it melting and refreezing of snow, deeply changes the
behaviour of snowpack from grain- to fluid-supported, making wet snow modelling much more complex than in case of dry
snow. Overall, partially saturated wet snow remains a critical challenge for modelling. In general, a snow cover modifies the
overall density and rigidity of the investigated medium, and thus the propagation velocity of seismic waves. Furthermore,
melt water runoff from the snowpack can percolate through the subsurface, increasing pore pressure and density of the
porous medium. Snowfall and snowmelt periods are therefore expected to affect seismic surface wave propagation, leading
to dV/V changes.
To better understand and constrain the effect of snow on dV/V, we deployed seismic sensors during an entire winter season
at a site in the Eastern Swiss Alps. We measured substantial dV/V changes related to snowfall and melting, indicating a

detectable effect of snow cover variations at this site. These observations were compared to theoretical values of dV/V computed from a mechanical model based on snow cover and subsurface elastic properties. Our results are of interest for

seismology, through a better interpretation of seismic measurements in snowy regions, and for snow cover monitoring, through the potential estimate of snowpack properties and their influence to subsurface by seismic measurements.

## 2 Field site and instrumentation

The seismic monitoring system was installed to monitor snow avalanches (Heck et al., 2019). It consisted of seven vertical geophones (Fig. 1b) with an eigenfrequency of 4.5 Hz, and data were recorded using a 24-bit acquisition system with a

sampling rate of 500 Hz (van Herwijnen and Schweizer, 2011). To increase the signal-to-noise ratio, the sensors were buried 30 to 50 cm deep as suggested by Heck et al. 2018. For this study, we used data from two sensors deployed at a distance of 35 m (yellow dots in Figure 1c). Data were collected from 17 December 2018 to 11 June 2019.

The instrumentation was deployed at the Jenatschalp field site in the Dischma valley above Davos (Eastern Swiss Alps; 46.73N, 9.91E; Fig. 1a). The field site is a flat meadow at an elevation of 1930 m a.s.l. surrounded by mountain peaks that

rise up to 3000 m. The field site was also equipped with seven automatic cameras installed at two different locations for visual snow thickness estimation of the site and the adjacent slopes.

## 3 Results of measurements

### 3.1 SNOWPACK simulations

To estimate snowpack properties at the location of the seismic sensors, we generated a one-dimensional snowpack

simulation using the snow cover model SNOWPACK (Lehning et al., 1999; Bartelt and Lehning, 2002). SNOWPACK simulates snow microstructure and the layering of the snowpack based on weather data. It is based on a Lagrangian finite element implementation and solves the non-stationary heat transfer and settlement equations. It encompasses phase transitions and the transport of liquid water. The model provides detailed information on the mechanical and physical properties of each snow layer, including temperature, density, liquid water content and snow microstructural descriptors. As

there were no meteorological measurements as input data at the site, we interpolated measurements from seven automatic weather stations (AWS) within a radius of 20 km of the field site at elevations ranging from to 1563 to 2558 m a.s.l. (Fig. 1a). All AWS provided half-hourly measurements of air temperature, relative humidity, wind speed and direction. Measured precipitation with a heated rain gauge as well as incoming short- and longwave radiation were only available at 2, respectively 3 AWS. For the spatial interpolations, , we used the preprocessing library MeteoIO (Bavay and Egger, 2014)

included in the SNOWPACK model. For most of the meteorological parameters, we used the IDW-LAPSE algorithm, which combines inverse distance weighting with a lapse rate.  To estimate the snow surface temperature, energy fluxes at the snow-atmosphere boundary were calculated (Neumann boundary conditions). For the soil heat flux at the bottom of the snowpack,

we set a constant value of 0.06 W/m2, which approximates the geothermal heat flux (Davies and Davies, 2010). The flow of liquid water through the snowpack was simulated using Richards equations (Wever et al., 2014). With the starting date set to 15 September 2018, the simulation was run with a time step of 15 min until all snow on the ground had melted on 7 June 2019. This melt-out data coincided well with the disappearance of the snow on the images of the automatic cameras.

To model the influence of the snowpack on changes in seismic velocities (see Sect. 4), we divided the entire snowpack in two layers with each a density and temperature equal to the depth-averaged density and temperature of all sub-layers. In winter, when the snowpack is cold and dry (i.e. snow temperature below 0°C), the two layers represent the settled base of the snowpack and the layer of fresh snow on top which is typically less dense (Figure 2). In spring, when the snowpack melts (i.e. snow temperatures at 0°C), the two layers represent the base of the snowpack that stays at 0°C, and the upper layer of the snowpack that periodically refreezes, for instance during the night or during cold weather. To define these two layers at each modelling time step we used the following procedure:

- In winter, we first determined the amount of new snow in the past 48 hours ($HN_{48}$, black line in Fig. 2). If $HN_{48} = 0$, then the entire snowpack consisted of one layer with a thickness equal to the snow depth HS. However, if $HN_{48} > 0$, we then determine the depth $d_{max}$ of the lowest layer within $HN_{48}$ consisting of precipitation particles or decomposed and fragmented particles (Fierz et al., 2009) and a density lower than 220 kg/m$^3$. For $d_{max} = 0$ the snowpack again consisted of one layer, while for $0 < d_{max} < HN_{48}$ the snowpack consisted of two layer with thickness HS-$d_{max}$ and $d_{max}$ (Figure 2).

- In spring, we determined the depth $d_{cold}$ of the lowest layer from the snow surface with a negative temperature. For $d_{cold} = 0$ the entire snowpack consisted of one layer with a thickness equal to the snow depth HS, while for $0 < d_{cold} < HS$ the snowpack consisted of two layer with thickness HS-$d_{cold}$ and $d_{cold}$.

## 3.2 Seismic observations

From raw seismic measurements, we derived dV/V by using the common method of ambient noise correlation (Campillo and Paul, 2003; Bensen et al., 2007; Larose et al., 2015). First, we pre-processed the 6-hour long raw seismic recordings by substracting the mean, detrending, clipping and spectral whitening between 0.2 and 30 Hz. We then calculated the cross-correlations of the two sensors with 3600 s long time windows, and applied a Wiener filter (with a 10 x 10 local window size, (Moreau et al., 2017)) to the resulting correlogram. From this filtered correlogram, we selected a time window from 0.2 to 0.5 s in both causal (correlation time >0) and acausal (correlation time <0) codas (Figure 3), which are known to be sensitive to elastic properties of the extended subsurface between sensors. In these time windows, we estimated the relative velocity change (dV/V) and the corresponding correlation coefficient (CC) by using the stretching method (Hadziioannou et al., 2011; Le Breton et al., 2021). We thus have dV/V time series with 4 values per day during the entire data period, in different frequency bands ranging from 10 to 25 Hz with a bandwidth of 4 Hz. Such seismic observations are shown in Figure 4. On this figure the reference period is chosen from January to February 2019, in order to select a long period with dry snow during the winter season as reference.

By comparing the seismic observations with modelled snow cover (Fig. 4a), in particular modelled new snow and runoff, we identified variations in dV/V and CC associated to snowfall and snowmelt periods, with different responses in intensity and frequency. We then decided to focus on the most significant periods during which a snow cover variation lead to a dV/V response: three snowfall events between 22 December and 15 January (respectively named SF0, SF1 and SF2), and two main snowmelt periods between 15 April and 29 May (respectively named SM0 and SM1). These periods are highlighted in Figure 4.

In order to quantify dV/V to snowpack variations accurately, for each of the three snowfall and two snowmelt periods we used new reference periods covering seven days before the start of the period of interest. In this case, dV/V are close to zero just before the event, and changes in dV/V are then expected to be related to variations in the snowpack. Such seismic observables are shown for each event, together with snow cover depth variations highlighting significant snowfalls or snowmelts (Fig. 5-9). When the correlation coefficient (CC) was too low (we fixed the minimal threshold arbitrarily at 0.6), we considered uncertainties in dV/V as too high, and removed the corresponding values. Since phase aliasing and cycle-skipping are known to occur using the stretching method (James et al., 2017), we also removed few dV/V outliers (singular values with more than 10% absolute difference with their neighbors) that should not be physically interpretable.

Overall, we observed a dV/V decrease for significant snowfall events (SF0, SF1 and SF2). For the earlier main snowfall (SF0), the decrease was minor (less than a percent, see Fig. 5). However, for the following snowfalls (SF1 and SF2), we observed decreases in dV/V of several % just after the event (Fig. 6-7), suggesting a more important role of fresh and dry snow in elasticity change of the surveyed medium than during SF0. In other words, additional fresh snow brings new mass onto the existing layer, without bringing any significant rigidity. Furthermore, the dV/V and CC responses were most sensitive in the frequency band around 20 Hz, for all cases.

For both melting periods (SM0 in Figure 8, and SM1 in Figure 9), we also observed a dV/V decrease of several %, especially for high frequencies (above 16 Hz). For SM0 there was a slight increase in dV/V for low frequencies (below 15 Hz). For SM1, changes in dV/V occurred over a longer time period, suggesting that the subsurface likely moistened or saturated during the melt-out phase of the snowpack, leading to a loss of rigidity.

Overall, these observations suggest that there is a substantial influence of the snowpack and ground subsurface below on seismic wave velocities. We address this quantitatively by a modelling step detailed in the following part.

## 4 Modelling

In this study we use the coda of cross-correlations from a pair of sensors at a distance of around 50 m, hence monitoring the subsurface through diffused surface waves. Thus, the dV/V measurements account for the variation in surface wave velocity. The following part aims to model such velocity before and after the periods of interest (snowfalls and snowmelt), accounting for elastic changes due to snowpack changes, in order to compare modelled dV/V variations to observed ones. To handle this

question, we built a physical model based on linear elasticity, with elastic parameters of the surveyed medium as inputs, which compute surface wave velocity along frequency.

Among environmental factors, we assume that snowpack changes play the major role leading to surface wave velocity fluctuations consecutive to snowfalls or snowmelt events. For example, atmospheric pressure changes may probably influence measured dV/V, but we expect the amplitude of this effect negligible (less than 0,1 % for a variation of few kPa) (Le Breton et al., 2021; Hotovec-Ellis et al., 2014).

Input parameters contain elastic (P-wave and S-wave seismic velocities) and inertial (density) properties of the medium, modelling the ground subsurface and the snow layers above. From this 1D model, the corresponding surface wave dispersion curve is then obtained as a result of the forward problem solved by the Geopsy package (Wathelet et al., 2004), using the linear theory of elasticity (Wathelet, 2005) and assuming that surface waves are mostly dominated by Rayleigh waves (Grêt et al., 2006). In fact, the energy partitioning dynamics favors Rayleigh waves in the early part of coda, when considering vertical component sensors and most of seismic noise sources being at (or almost) the surface (Obermann et al., 2013). Moreover, it is worth noticing that our study do not depend on the depth of geophones, since we studied only surface wave phase velocities that are not depth-dependent (contrarily to the wave amplitude). We then estimate Rayleigh wave velocities just before and just after the event (snowfall or snowmelt), allowing to deduce the modelled relative velocity variations (dV/V) along frequency, for each event.

### 4.1 Numerical ground parameterization

To model surface wave propagation within the ground subsurface, we performed P- and S-wave refraction surveys in July 2020, employing 24 geophones (horizontal and vertical) and sledgehammer strikes (Fig. 10).

Assuming a horizontal layered medium (which, from geological and geomorphological studies, is partially true), we deduced from time-distance plots of the first arrivals a three layers model down to a depth of about 20m. Note that, as usual, the P-wave profile goes deeper than the S-wave profile, the latter not allowing to resolve the second interface at around 15 m depth.

The first layer (0-1m) consists of vegetated clayey drained moraine (Vp = 470 +/- 50 m/s, Vs = 110 +/- 20 m/s, estimated density $\rho$ = 1500 +/- 150 kg/m$^3$), overlaying a similar layer with less organic content (1-2.3m, Vp=470 +/- 50 m/s, Vs= 800 +/- 80 m/s, est. density $\rho$ =2300 +/- 200 kg/m$^3$). Then, the water table is reached in a morainic terrain (2.3-17 m : Vp=1500 +/- 100 m/s, Vs=800 +/- 80 m/s). Below 18 m, the bedrock is likely constituted of consolidated crystalline rocks (Vp=3900 +/- 200 m/s, est. density $\rho$ =2500 +/- 200 kg/m$^3$). In that latter unit, we estimate the shear wave velocity (Vs=2100 +/- 150 m/s) assuming a Poisson's ratio of 0.25-0.30 (Tarkov and Vavakin, 1982) which are average values for consolidated rocks.

Densities were estimated from the literature (Taylor and Blum, 1995) and the geological map, keeping in mind that densities have limited variations for different lithologies and feebly impact surface wave velocity variations. Also, considering the frequency of the surface waves studied here (mainly between 10 and 25 Hz), bedrock seismic

parameters play a limited to negligible role, such that it was not necessary to obtain better estimations below 17 m depth. All parameters of the ground model are summarized in Table 1. We also assumed that these ground parameters are unchanged during all the season.

Table 1 : Numerical ground model deduced from geophysical investigations. These parameters are used in order to model dV/V values, and they are assumed constant before and after snowfall events.

|  | Vp (m/s) | Vs (m/s) | Poisson's ratio | Density (kg/m³) | Thickness(m) |
|---|---|---|---|---|---|
| Vegetalized soil | 470 | 110 | 0.47 | 1500 | 1 |
| Top moraine | 500 | 300 | 0.22 | 2300 | 1.3 |
| Moraine | 1500 | 800 | 0.30 | 2300 | 14.7 |
| Bedrock | 3900 | 2100 | 0.30 | 2500 | ∞ |

## 4.2 Numerical ground parameterization

Snowpack properties were estimated from modeled density and temperature of each layer (see Sect. 3.1). Seismic parameters are then computed by using empirical relations for Vp and Vs, assuming a Poisson's ratio of the snow equal to 0.3. This modelling step deals only with dry snow, since no liquid water is taken into account for the sake of simplicity.

First we address the relationship between snow density and Young's modulus $E$ at a reference temperature $T_{ref} = -5°C$
(Gerling et al., 2017):

$$E_{ref}(\rho) = 6.10^{-10}.\rho^{4.6} \tag{1}$$

In parallel we use the temperature-Young's modulus relation with $T_m = 273\ K$ and $E_0 = 0.75\ MPa$ the reference shear modulus measured at 263 K (Schweizer and Camponovo, 2002) :

$$\ln\left(\frac{E}{E_0}\right) = A_0 + A_1\exp\left[\alpha_1\left(\frac{1}{T} - \frac{1}{T_m}\right)\right] + A_2\exp\left[\alpha_2\left(\frac{1}{T} - \frac{1}{T_m}\right)\right] = f(T) \tag{2}$$


with :

$A_0 = 0.747, A_1 = -1.24, \alpha_1 = -3.85.10^3 K, A_2 = -6.45, \alpha_2 = -1.82.10^5 K$.

By combining these two expressions (1) and (2), we obtain a temperature and density dependent Young's modulus for snow:

$$E(\rho, T) = E_{ref}(\rho) \frac{\exp\left(f(T)\right)}{\exp\left(f(T_{ref})\right)} \tag{3}$$


Seismic velocities are then deduced as follows (classical formula):

$$V_p = \sqrt{\frac{E(1-\nu)}{\rho(1+\nu)(1-2\nu)}} \tag{4}$$

with a Poisson's ratio of snow $\nu = 0,3$, and from (Capelli et al. 2016, Fig.1):

$$V_s \approx \frac{1}{2} V_p \tag{5}$$

We then obtained snow models for the three snowfall events (SF0, SF1, SF2), before and after the main increase in snow depth. We also apply a model for the first melting period (SM0) before and after the observed dV/V perturbation. The results of this parametrization step are summarized in Table 2-5, respectively.

**Table 2 : Values of the snow model for snowfall 0 (SF0).**

|  | Before snowfall 0 (23.12.2018) | | | | After snowfall 0 (25.12.2018) | | | |
|---|---|---|---|---|---|---|---|---|
|  | Vp (m/s) | Vs (m/s) | Density (kg/m3) | Thickness (cm) | Vp (m/s) | Vs (m/s) | Density (kg/m3) | Thickness (cm) |
| Top snow | 220 | 110 | 170 | 2 | 300 | 150 | 180 | 23 |
| Bottom snow | 450 | 225 | 240 | 53 | 600 | 300 | 260 | 51 |

**Table 3: Values of the snow model for snowfall 1 (SF1).**


|  | Before snowfall 1 (01.01.2019) | | | | After snowfall 1 (03.01.2019) | | | |
|---|---|---|---|---|---|---|---|---|
|  | Vp (m/s) | Vs (m/s) | Density (kg/m3) | Thickness (cm) | Vp (m/s) | Vs (m/s) | Density (kg/m3) | Thickness (cm) |
| Top snow | 160 | 80 | 130 | 4 | 150 | 75 | 120 | 20 |
| Bottom snow | 600 | 300 | 260 | 68 | 640 | 320 | 260 | 70 |

**Table 4: Values of the snow model for snowfall 2 (SF2).**

|  | Before snowfall 2 (13.01.2019) | After snowfall 2 (15.01.2019) |
|---|---|---|

| | Vp (m/s) | Vs (m/s) | Density (kg/m3) | Thickness (cm) | Vp (m/s) | Vs (m/s) | Density (kg/m3) | Thickness(cm) |
|---|---|---|---|---|---|---|---|---|
| Top snow | 130 | 65 | 150 | 8 | 240 | 120 | 150 | 50 |
| Bottom snow | 600 | 300 | 250 | 110 | 650 | 325 | 270 | 120 |

**Table 5 : Values of the snow model for snowmelt 1 (SM0).**

| | Before snowmelt 1 (22.04.2019) | | | | After snowmelt 1 (25.04.2019) | | | |
|---|---|---|---|---|---|---|---|---|
| | Vp (m/s) | Vs (m/s) | Density (kg/m3) | Thickness (cm) | Vp (m/s) | Vs (m/s) | Density (kg/m3) | Thickness (cm) |
| Homogeneous snow | 60 | 30 | 460 | 116 | 60 | 30 | 460 | 96 |


In brief, we summarize both instrumentation of the site and 1D modelling protocol by a schematic cross-section for the snowfall event 2 (Figure 11).

## 4.3 Results of modelling

We computed Rayleigh wave propagation velocities by Geopsy, for each model composed of stacked snow and ground
layers (see Table 1 for ground and Tables 2-5 for snow), before and after each snowpack event. The relative velocity change between the model before and after the event was then considered as the modelled dV/V values, which are computed for different frequency bands.

Then we compared observed and modelled values of dV/V with frequency (Figure 12a for SF0, Figure 12b for SF1, Figure 12c for SF2, Figure 12d for SM0). Model results are shown with errorbars corresponding to snow elastic parameters
uncertainties (P- and S-wave velocities +/- 10%), in order to assess the sensitivity of the model to snow modelling.

For all the three snowfall events (SF0, SF1, SF2), both observed and modelled dV/V are in the same order of magnitude, reinforcing the interpretation of changes in dV/V as a response to snow depth increase. Nevertheless, modelled dV/V were generally over-estimated for SF0 event, where only very small dV/V variations were observed. In this case, the sensibility of dV/V measurements reaches probably its limits for this snowfall. For SF1 and SF2 events, however, the model is in good
agreement with observations.

In contrast, our model did not match with observations for SM0 event. Modelled dV/V are positive with very high values, whereas we observed negative dV/V. It is worth noticing that our model assumes a totally dry snow when estimating elastic properties. But the moistening of snowpack and shallow ground layers below is a common process occurring in early and late spring, probably changing the elastic behaviour of the snowpack during this melting period due to the presence of liquid
water. Nevertheless, Figure 12d shows the limit of validity of our model, that address only a dry medium (snowpack and ground) in early winter season.

## 5 Discussion

In this study we measured changes in dV/V at a snow covered site over an entire winter season. We modelled the results with relatively good agreement, except during snowmelt. This modelling aims at assessing the effect of snowpack variations on dV/V measurements. We reproduced dV/V decrease due to a snowfall, with the same order of magnitude than the observed values. Some uncertainties are still unclear, and may explain the gap between observed and modelled values. Uncertainties of elastic parameters of the snowpack are mentioned above. For the ground subsurface, the sensitivity of the model is negligible for deep layers, so that bedrock uncertainties do not play any role here. However, the model is more sensitive to elastic parameters of shallow layers, especially S-wave velocity since we assume to monitor Rayleigh surface waves. Hence, the uncertainties linked to the shallow layers of the ground may induced errors in the results. The sensitivity of our model to snow elastic properties was addressed by accounting for +/- 10% variations, resulting in modelled dV/V that can vary by several percent (see Fig. 12), especially for high frequencies (above 15 Hz). Finally, our physical model based on surface wave propagation velocity may be improved by considering the effect of liquid water on the noise wavefield and its changes in frequency content, that is recorded by buried seismic sensors along the season, in a view of detecting spurious dV/V estimates.

For the three snowfall periods (SF0, SF1, SF2), the agreement between observed and modelled values of dV/V reinforce our interpretation : a snowfall event has a substantial and almost direct effect on dV/V measurements, with a decrease of several percent in a frequency band between 15 and 25 Hz at our site. Since we consider fresh and dry snow, this decrease is probably related to an increase of the overall mass of the surveyed medium induced to the additional snow weight several hours after a snowfall event, without rigidity increase (since fresh snow has little rigidity).

For melting periods (SM0), our model was not able to reproduce the observations, probably because of the significant change in elastic behaviour induced by liquid water percolation into the snowpack and the subsurface. The parameterizations used for the elastic properties of snow were based on laboratory measurements of dry snow (Schweizer and Camponovo, 2002; Gerling et al., 2017). However, we apply those to a wet snowpack, and therefore do not account for the influence of liquid water in the snowpack. To better model the influence of liquid water in both the snow and ground, a poro-elastic three-phase approach is likely required to accurately estimate elastic parameters (especially for realistic Vp and Vs values) (Sidler, 2015), but that is out of the scope of this article. At the most, we can expect that the presence of liquid water increases the density, and melting decreases the rigidity (contacts between grains), all together decreasing the shear wave velocity, and thus decreasing dV/V (Grêt et al., 2006; Voisin et al., 2017, 2016; Sidler, 2015).

Moreover, not every snowfall event led to a clear dV/V response during the entire winter season (Fig. 4). In our case, only three snowfall periods show a substantial effect on seismic velocities, suggesting that this snow effect is relative. Indeed, it depends on elastic parameters gap between snow layer and underlying ground layers: if the density of new snow is not that much different than the existing snowpack (for dry snowpack in early winter, as SF0), or if the additional new snow layer is negligible compared to the entire snowpack thickness (for thick and compacted snowpack in late winter, as March), this

effect will be minor and less detectable. These latter statements have been confirmed by our model : fresh dry snow on compacted snowpack has little influence on dV/V (modelled variation less than 1% in the considered frequency band).

For early snowfalls (SF0, SF1, SF2), these observations demonstrate that the dV/V is well modeled by surface wave phase velocity changes due to the successive snow layers, providing that the elastic properties of each layer is properly independently estimated. Improving the fits of both seismic and snowpack time series presented in the study requires more refined field observations or small scale mechanical models. As a long term perspective of the present work, dV/V will be used to better assess the mechanical properties of the snow layers, with a time resolution below the day and uncertainties below 10%.

## 6 Conclusion

We addressed the effect on snowfall and snowmelt on seismic velocity variations, derived from ambient noise correlation. From observations over a winter season, we actually measured dV/V drops related to snowpack thickness changes. We modelled these dV/V decreases by elastic changes in dry snowpack, that explains well the observed values. When a snowfall brings a new fresh snow layer that significantly differs from the medium below in terms of rigidity and density, it induces elastic changes measurable by a pair of seismic sensors. Finally, the present study gives a quantitative knowledge of the snow effect on dV/V : this response can be inverted to finer constrain mechanical properties of a snowpack, while the interaction between snowpack and subsurface has to be addressed for an accurate seismic monitoring in snowy regions.

## 7 Acknowledgments and data availability

This work was supported by the ANR LABCOM GEO3ILAB project, and was partly funded by the WSL research program Climate Change Impacts on Alpine Mass Movements – CCAMM (ccamm.slf.ch). Snowpack simulations and seismic data are available upon request at WSL Institute for Snow and Avalanche Research SLF (Davos, Switzerland).

## 8 Author contribution

AH designed and supported the meteorological and seismic experiments on the Jenatschalp site. EL and AH performed the active seismic refraction survey on the study site. Seismic data processing and mechanical modelling have been developed by AG, in close collaboration with LB, EL and AH. SNOWPACK simulations have been designed and processed by SM. AG prepared the manuscript with contributions from all co-authors.

## 9 Competing interests

The authors declare that they have no conflict of interest.

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

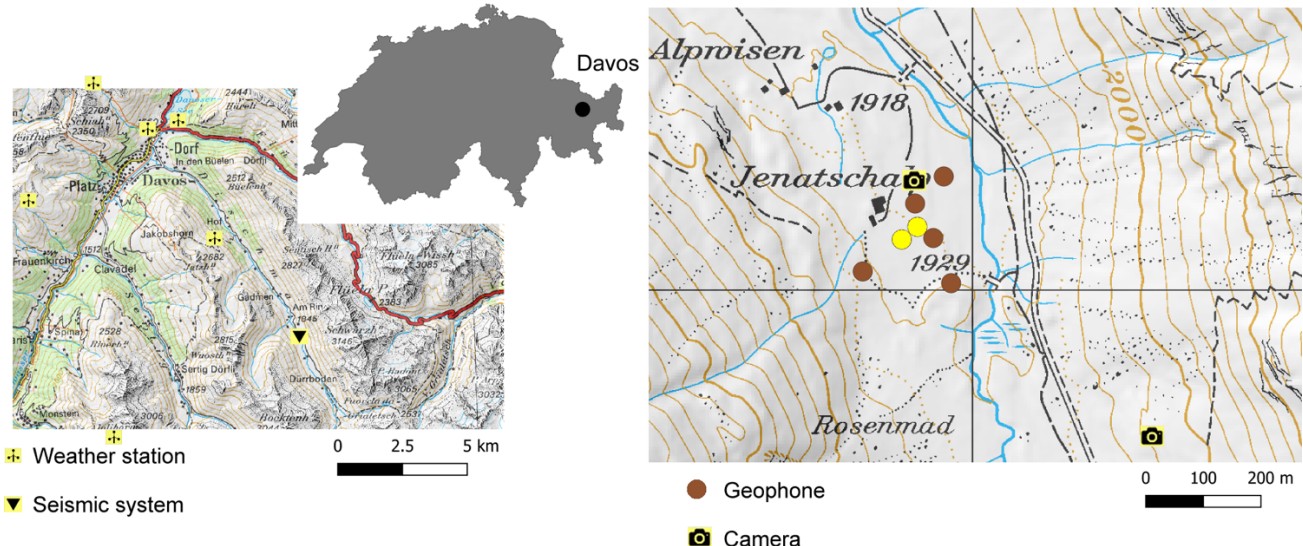

**Figure 1:** (a) **Map of the Davos area, Switzerland. The location of the seismic system is shown by the black triangle, the wind wheel shows the locations of 6 of the 7 the weather stations that provided input data for SNOWPACK. (b) Detailed map of the Jenatschenalp site showing the geometry of the seismic array and the positions of automatic cameras. The yellow circles indicate the positions of sensors used in this study. Reproduced with permission from Swisstopo (JA100118).**


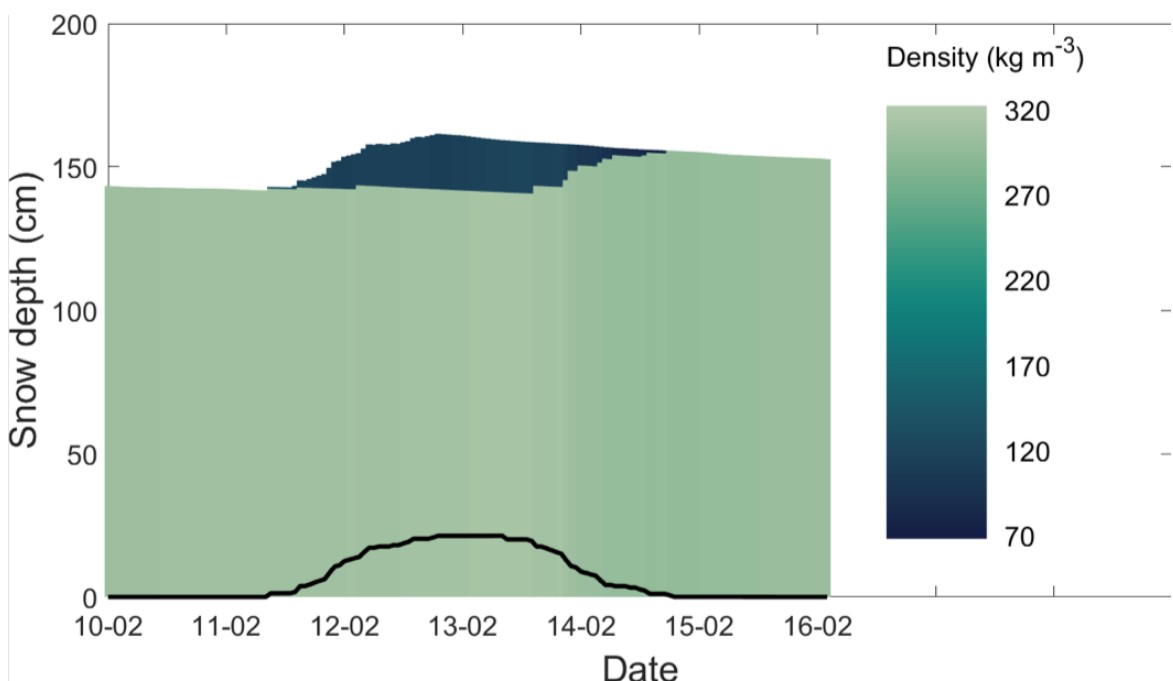

**Figure 2: Evolution of snow density (colors) of the simplified snowpack consisting of two layers, during one snowfall event. When HN48 (black curve) was zero, both layers have the same density. For HN48 > 0, the upper layer consists of lower density snow (dark blue).**

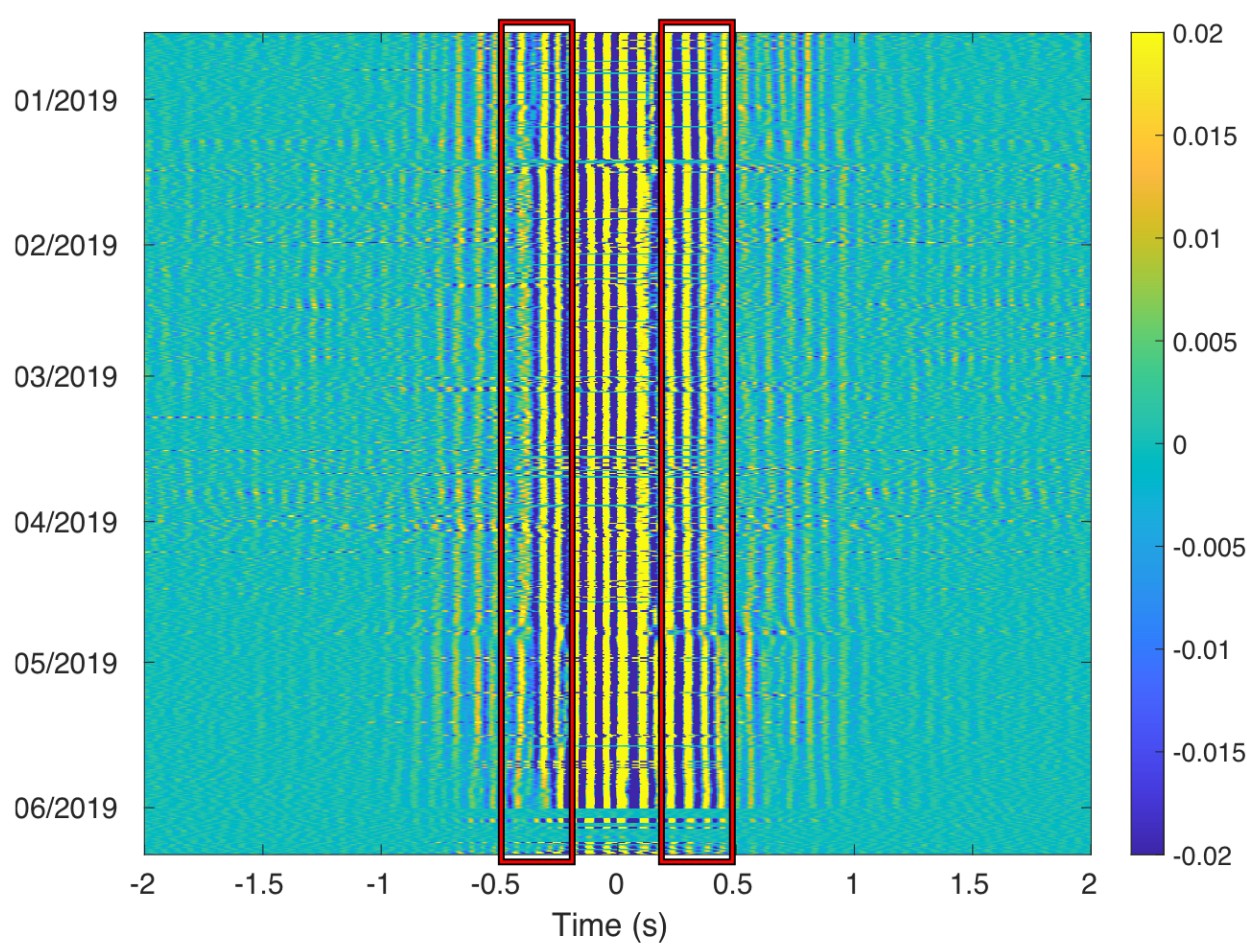


**Figure 3: Filtered normalized correlogram from raw seismic noise cross-correlations over the pair of geophones used for the study. The time windows from which the dV/V values are estimated are localized by red boxes, corresponding to direct (positive) and indirect (negative) coda part of the waveforms.**

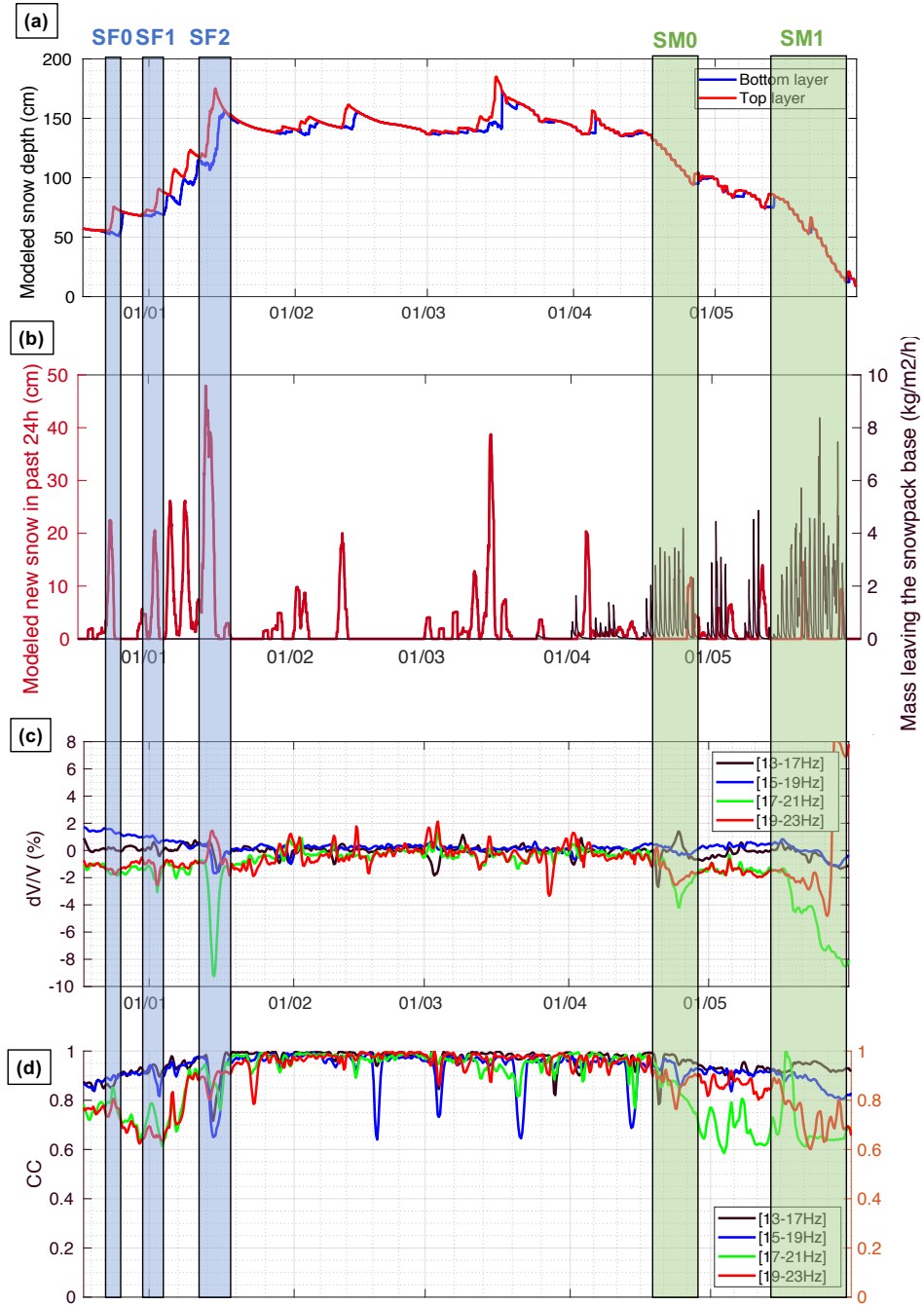

**Figure 4: Results of snow simulations over the entire season from December 2018 and June 2019, with (a) interpolated snow depth of layers defined by a procedure based on density, and (b) modeled new snow in past 24 hours (in red) and mass leaving the snowpack base, highlighting melting in spring (black curve). Seismic observations are also presented over the same period, with relative surface wave velocity changes (dV/V) (c) and correlation coefficient (CC) (d) for different frequency bands (see legend). From these time series, we select three snowfall events (SF0, SF1, SF2 in blue boxes) and two melting periods (SM0 and SM0 in green boxes), during which a significant and simultaneous dV/V response occurs.**

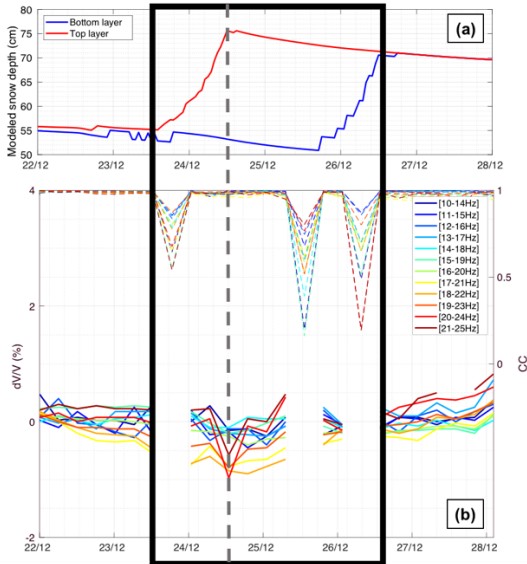

**Figure 5: Observations during the snowfall event 0 (SF0), with modeled depth of each snow layer from Snowpack simulations (a), and dV/V measurements at different frequency bands, and corresponding CC values in dashed lines (b). When the correlation coefficient (CC) was too low (CC < 0.8), dV/V values are considered as outliers, and then removed. The frame in black shows approximately the whole period of interest, whereas the grey dotted line highlights precisely the state of the medium with corresponding observables, just after the event.**

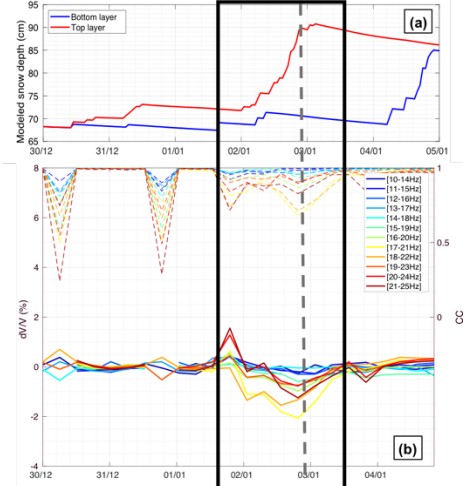

**Figure 6: Observations during the snowfall event 1 (SF1), with modeled depth of each snow layer from Snowpack simulations (a), and dV/V measurements at different frequency bands, and corresponding CC values in dashed lines (b). When the correlation coefficient (CC) was too low (CC < 0.6), dV/V values are considered as outliers, and then removed. The frame in black shows approximately the whole period of interest, whereas the grey dotted line highlights precisely the state of the medium with corresponding observables, just after the event.**

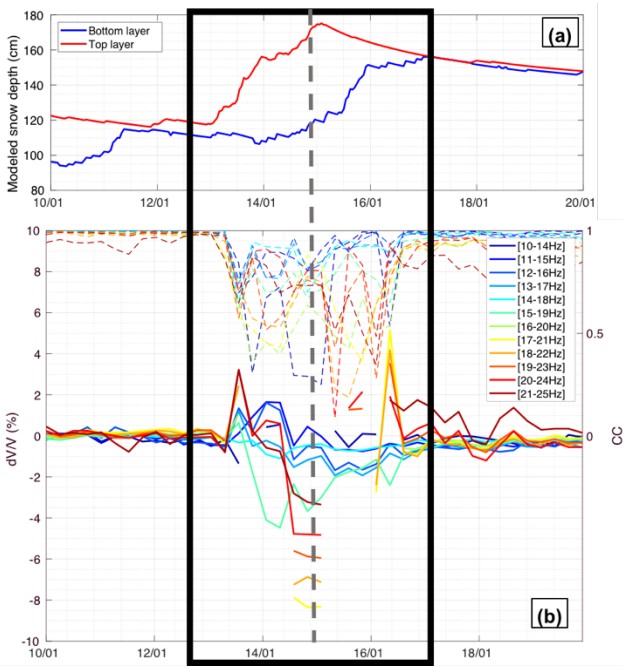

**Figure 7: Same legend as Figure 6, for snowfall event 2 (SF2).**

470

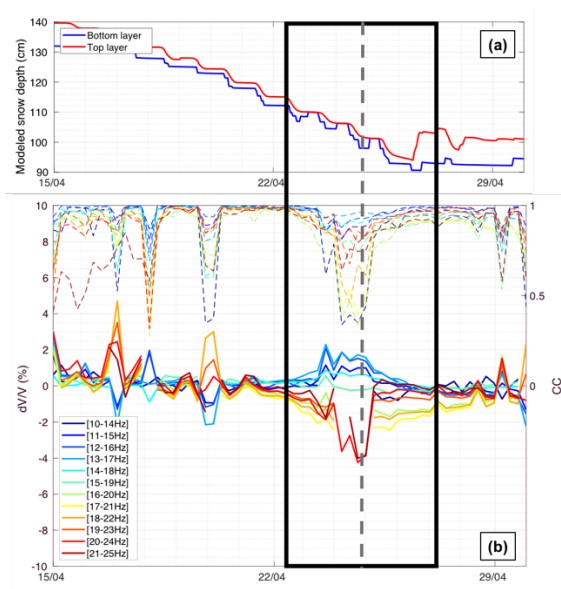

**Figure 8: Same legend as Figure 6, for snowmelt event 0 (SM0).**

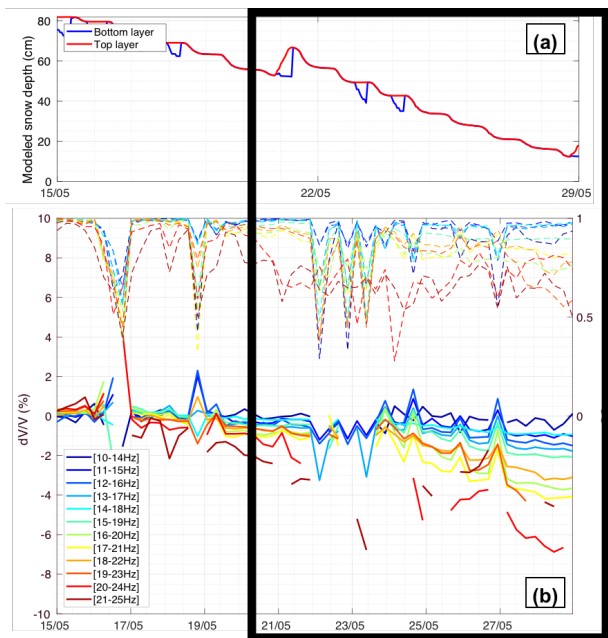

**Figure 9: Same legend as Figure 6, for snowmelt event 1 (SM1).**

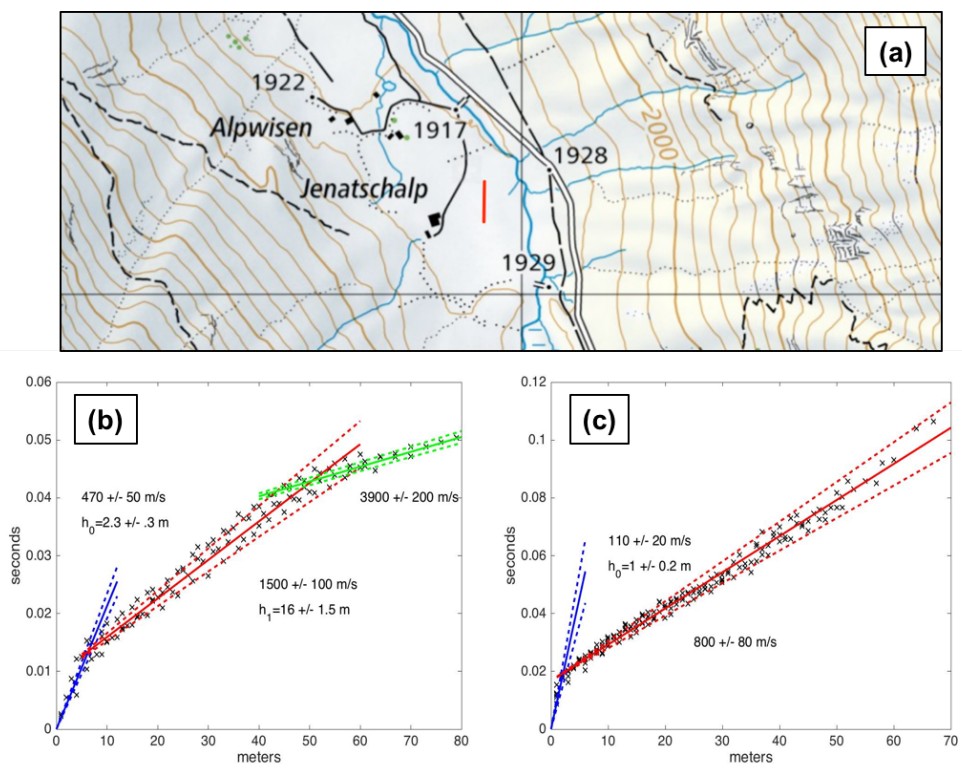

Figure 10: (a) Location map of the geophysical investigations in Jenatschalp site (red profile). (b) Results of the active seismic refraction for P-wave velocity (Vp) layering (b) and S-wave velocity (Vs) layering (c).

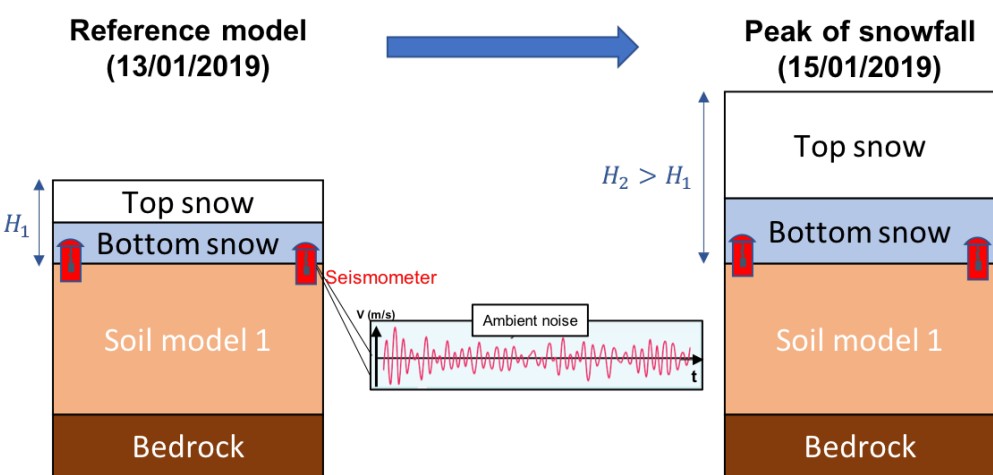

**Figure 11 : Schematic 1-D cross-section of the instrumentation of the study site, with the location of seismic sensors buried in shallow subsurface, and the modelled layered medium at two temporal steps (before and during peak of snowfall event 2, as an example). The only changes between these models is the increasing snow depth and mechanical properties of both snow layers, as precised in Table 4.**

480

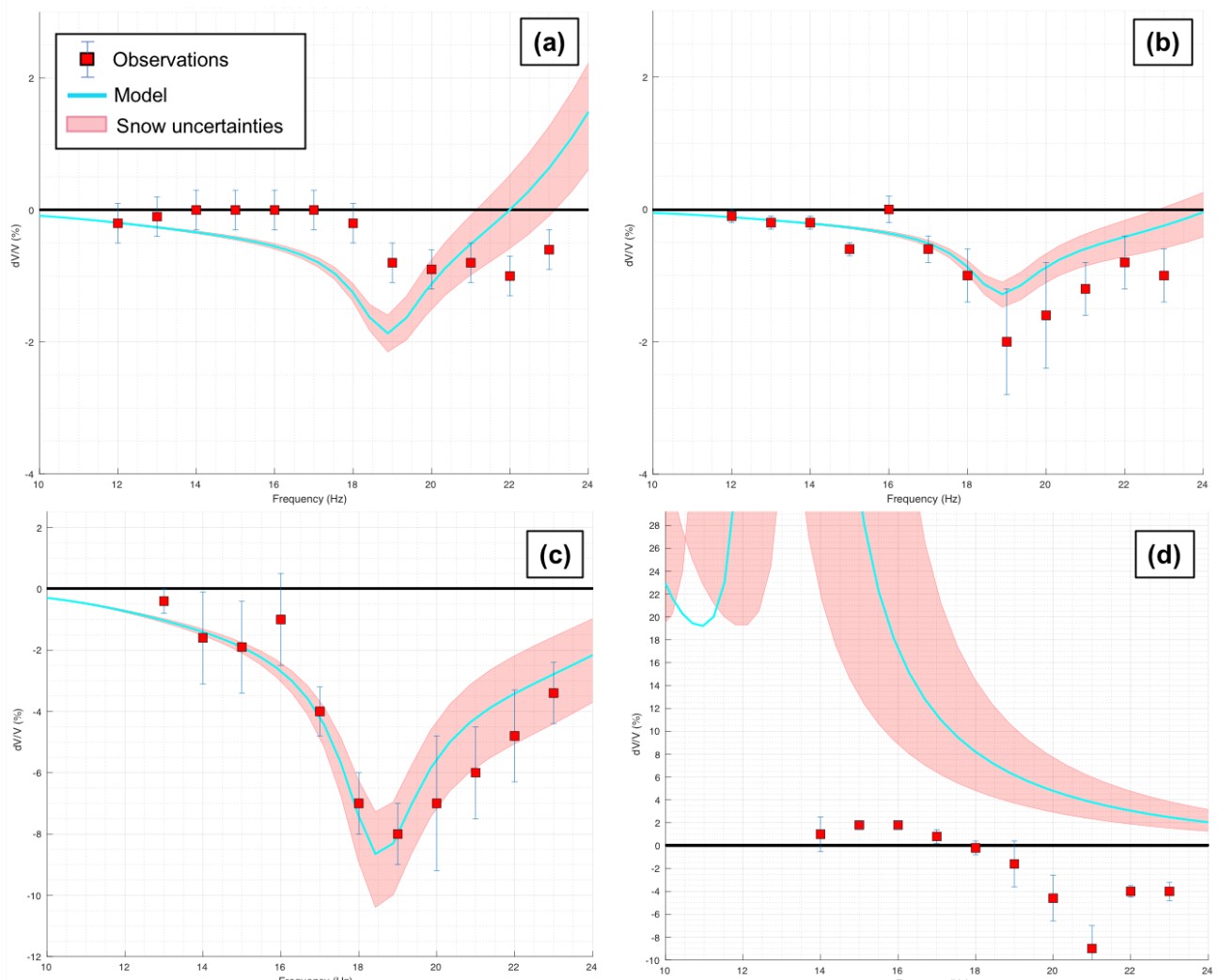

**Figure 12: (a) Results of the dV/V modelling for snowfall event 0 (SF0), with modeled dV/V response with respect to frequency (blue curve) and uncertainties (shaded pink curves) related to +/- 10% variations in snow elastic parameters. Observations are highlighted in red squares, which frequency is fixed to the center of the frequency band of the measured dV/V. (b) Same legend for snowfall event 1 (SF1). (c) Same legend for snowfall event 2 (SF2). (d) Same legend for snowmelt event 0 (SM0).**

485