# Peer review of "Effect of snowfall on changes in relative seismic velocity measured by ambient noise correlation"

_The Cryosphere, 2021_

## Author Comment (AC1)

[Figure]

*Figure 1 : Normalized correlogram from raw seismic noise cross-correlations over the pair of geophones used for the study. The time windows from which the dV/V values are estimated are localized by red boxes, corresponding to direct (positive) and indirect (negative) coda part of the waveforms.*

[Figure]

*Figure 2 : Schematic 1-D cross-section of the instrumentation of the study site, with the location of seismic sensors buried in shallow subsurface, and the modelled layered medium at two temporal steps (before and during peak of snowfall event 2, as an example). The only changes between these models is the increasing snow depth and mechanical properties of both snow layers, as precised in Table 4.*

---

## Author Comment (AC2)

**Reference model
(13/01/2019)**

**Peak of snowfall
(15/01/2019)**

Top snow

$H_1$

Bottom snow

Seismometer

Soil model 1

Bedrock

$H_2 > H_1$

Top snow

Bottom snow

Soil model 1

Bedrock

v (m/s)

Ambient noise

t

*Figure 1 : Schematic 1-D cross-section of the instrumentation of the study site, with the location of seismic sensors buried in shallow subsurface, and the modelled layered medium at two temporal steps (before and during peak of snowfall event 2, as an example). The only changes between these models is the increasing snow depth and mechanical properties of both snow layers, as precised in Table 4.*

[Figure]

*Figure 2 : Results of the dV/V modelling for snowfall event 2 (SF2), with modeled dV/V response with respect to frequency (blue curve) and observations highlighted in red squares, which frequency is fixed to the center of the frequency band of the measured dV/V. For modelling the snowpack , we used a 10 cm resolution (depth-averaging temperature and density profiles with 10 cm thick sub-layers).*

---

## Author Response (AR1)

**Effect of snowfall on changes in relative seismic velocity measured by ambient noise correlation**

Submitted in The Cryosphere

**Review 1 : Referee comments**

**Referee 1 :**

**General Comments**

The article identifies relative changes in subsoil stress caused by the snow cover in its fresh and dry state (when it is melting). The authors use ambient seismic noise to calculate these changes using coda DV/V wave interferometry. The hypothesis is that the melting snow can percolate through the soil surface and increase the pore pressure and density, leading to possible mass slips.

The article is well structured and adequately written. A significant contribution is that experimental results can be correlated with numerical simulations, which show that relative stress changes can be reproduced for the two physical states of the snow cover.

The article can be accepted with minor corrections.

I suggest a discussion of variations in dV/V estimation if **atmospheric effects** are taken into account.

Such atmospheric effect might probably influence the measured dV/V where atmospheric change occurs, but we expect its amplitude negligible for most cases, compared to other environmental influences (Le Breton et al., 2021; Hotovec-Ellis et al., 2014). The additional loading from the snow cover is much more important than atmospheric pressure variations. Then, following previous literature, we can argue that a dV/V variation less than 0,1 % for atmospheric changes (few kPa) is expected. We can add this discussion in the beginning of the part Modelling : "Among environmental factors, we assume that snowpack changes play the major role leading to surface wave velocity fluctuations consecutive to snowfalls or snowmelt events. For example, atmospheric pressure changes may probably influence measured dV/V, but we expect the amplitude of this effect negligible (less than 0,1 % for a variation of few kPa) (Le Breton et al., 2021; Hotovec-Ellis et al., 2014)."

Also, the authors should include a **figure showing the correlations** obtained and indicating the part of the waveform in which the dV/V estimation is made.

 Yes, we agree with this suggestion. Find the new figure in the new version of manuscript (Figure 2) showing the entire correlogram from seismic data, including the part of the waveform from which dV/V is estimated.

---

## Referee Report (RR1)

Dear Authors,

You have addressed all my questions and concerns from the first round of review – I sincerely thank you for that and find your manuscript much improved. I have some further minor comments:

Introduction

Some further elaboration is worthwhile. Readers need to understand why modelling snow is challenging, and why wet snow is even more so. Sayers 2021 models with diff. effective medium scheme the Vp and Vs dry snow using two phases: ice and air. If one were to model wet snowpack, one would further need to consider effects of partial saturation, another can of worms (for example O'Connell and Budiansky 1974 looked at partial saturation). In addition there would be critical conditions whereby snow would change its behaviour from grain-supported to fluid-supported. It is important to explain the complexity of snow in this regard, and the challenge it presents itself to theoretical models (as you mention in discussion the 3-phase model)

Line 103-113

Would benefit from a diagram, showing layers with properties A/B vs. layers with identity settled/fresh (which I understand from your description that they are different). Do I understand correctly that some of the HN48 has properties of A and some of B?

Line 263 increase in mass and also density?

I propose minor changes to your submission.

Congratulations on your manuscript.

---

## Author Response (AR2)

**Effect of snowfall on changes in relative seismic velocity measured by ambient noise correlation**

Submitted in *The Cryosphere (October 2021)*

Dear Editor, dear Referee,

Please would you find below a point-by-point response to the last review of our manuscript. Our contribution is highlighted in green, in response to the two remaining comments that we had to address. We hope that the whole process is complete before the final publication, and we thank you again for all the relevant comments on our work.

Best,

Antoine Guillemot

**Minor revision : comments from referee**

Dear Authors,
You have addressed all my questions and concerns from the first round of review – I sincerely thank you for that and find your manuscript much improved. I have some further minor comments:

**Introduction**

Some further elaboration is worthwhile. Readers need to understand why modelling snow is challenging, and why wet snow is even more so. Sayers 2021 models with diff. effective medium scheme the Vp and Vs dry snow using two phases: ice and air. If one were to model wet snowpack, one would further need to consider effects of partial saturation, another can of worms (for example O'Connell and Budiansky 1974 looked at partial saturation). In addition there would be critical conditions whereby snow would change its behaviour from grain-supported to fluid-supported. It is important to explain the complexity of snow in this regard, and the challenge it presents itself to theoretical models (as you mention in discussion the 3-phase model).

- To this end, we add these new sentences to the introduction about snow: "*Modelling snow acoustics is highly challenging, since acoustic phase velocities of this porous medium strongly depends on porosity, stiffness and density of the bulk frame. Recent studies address this dependency using rigid-frame and Biot's models, assuming pore space to be air-filled (Capelli et al., 2016; Sidler, 2015; Sayers, 2021). Furthermore, the presence of liquid water, and with it melting and refreezing of snow, deeply changes the behaviour of snowpack from grain- to fluid-supported, making wet snow modelling much more complex than in case of dry snow. Overall, partially saturated wet snow remains a critical challenge for modelling.*"

**Line 103-113**

Would benefit from a diagram, showing layers with properties A/B vs. layers with identity settled/fresh (which I understand from your description that they are different). Do I understand correctly that some of the HN48 has properties of A and some of B?
Line 263 increase in mass and also density?

- For the winter season, we compute the amount of new snow in the past 48 hours (HN48). If HN48 is positive, we model the snowpack with two layers (fresh snow and settled snow), with different density and temperature. According to your suggestion, we clarify this procedure by adding a diagram (Figure 2 in the article, Figure 1 here):

[Figure]

*Figure 1: Evolution of snow density (colors) of the simplified snowpack consisting of two layers, during one snowfall event. When HN48 (black curve) was zero, both layers have the same density. For HN48 > 0, the upper layer consists of lower density snow (dark blue).*

**References**

Capelli, A., Kapil, J. C., Reiweger, I., Or, D., and Schweizer, J.: Speed and attenuation of acoustic waves in snow: Laboratory experiments and modeling with Biot's theory, 125, 1–11, https://doi.org/10.1016/j.coldregions.2016.01.004, 2016.
Sayers, C. M.: Porosity dependence of elastic moduli of snow and firn, 1–9, https://doi.org/10.1017/jog.2021.25, 2021.
Sidler, R.: A porosity-based Biot model for acoustic waves in snow, 61, 789–798, https://doi.org/10.3189/2015JoG15J040, 2015.